# Landscape Function Analysis: Responses to Bush Encroachment in a Semi-Arid Savanna in the Molopo Region, South Africa

**Klaus Kellner [1,\*], Jaco Fouché [1], David Tongway [2], Ricart Boneschans [1], Helga van Coller [1,3] and Nanette van Staden [1]**

[1]  Unit for Environmental Sciences, School of Biological Sciences, Management at the North-West University, Private Bag X6001, Potchefstroom 2520, South Africa; jacobus.fouche1@gmail.com (J.F.); 21548005@nwu.ac.za (R.B.); h.vancoller@saeon.nrf.ac.za (H.v.C.); nanette.van.staden@gmail.com (N.v.S.)

[2]  School of Engineering, Science and Environment, Newcastle University or 14 Coppin Place, Weetangera, ACT 2614, Australia; dtongway@iinet.net.au

[3]  South African Environmental Observation Network, Ndlovu Node, Scientific Services, Kruger National Park, Private Bag X1021, Phalaborwa 1390, South Africa

\*  Correspondence: klaus.kellner@nwu.ac.za; Tel.: +27-(018)-299-2510 or +27-(0)825696145

**Abstract:** Various factors lead to increased woody species density, biomass and cover (so-called 'bush encroachment') that influence ecosystem functioning and services in semi-arid rangelands. Ultimately, bush encroachment has adverse effects on human livelihoods. An increased understanding of ecosystem functioning in bush-encroached rangelands could contribute to improved management, conservation and restoration. This study, therefore, aimed to determine landscape functioning of bush-encroached and controlled savanna rangelands in the Molopo region, South Africa, by using the landscape function analysis (LFA) monitoring procedure. Mixed models revealed no significant differences based on LFA indices between bush-thickened and bush-controlled sites due to drought conditions that prevailed while the survey was carried out. Stability, which revealed the largest LFA contributing factors, always had the highest numerical value for sites that were still bush-encroached. Soil analyses revealed that grass litter patches from aeroplane-controlled sites had the highest average nutrient levels. As expected, high percentages of carbon and calcium levels were found in bush-encroached shrub litter patches. Bush-encroached landscapes are fully functional areas, especially under drought conditions. Long-term research is required to determine the effects successful management has on ecosystem functioning, especially during periods of higher rainfall.

**Keywords:** bush control; ecosystem functioning; landscape functional analysis; patch and inter-patch zones; nutrient cycling; infiltration; rangeland management; stability

## 1. Introduction

The most prominent biome in South Africa is the savanna, which is utilised extensively for rangeland management in the production and conservation of livestock and wildlife, as well as for subsistence crop farming [1,2]. Vegetation structure and composition of savanna landscapes vary considerably due to diverse topography, soil and climate [3]. Additionally, land use and the management thereof can contribute to compositional and structural changes across such landscapes [4–6]. In response to overgrazing, combined with severe drought events and climate change, savanna rangelands can be subjected to degradation that leads to an increase in the density of woody species (trees and shrubs), also called bush encroachment [7–11].

In areas where woody species co-exist with grasses and their density increases, bush encroachment is often referred to as 'bush thickening', as the woody species 'thicken' causing bush encroachment [12]. Note that in this paper, bush-thickened sites (BT) will be considered and described as bush-encroached sites. Bush encroachment causes shifts in vegetation types, e.g., from open savanna to closed woodland [13]. Although the main

causes of bush encroachment (which includes bush thickening) are poorly understood and widely debated [8,14], this phenomenon has adverse effects on the environment which, among other factors, results in a decrease in perennial grass production and soil erosion leading to a lack of sufficient surface cover [13,15,16].

Ref. [17], as well as Ref. [18], found that bush encroachment can occur within a decade depending on factors such as land use and management, climate, fire and the success of follow-up treatments to curb the recruitment and successful establishment of saplings [7,19]. Although trees and grasses have coexisted in savanna rangelands for thousands of years [20,21], increased densities of woody species outcompete the herbaceous layer for resources [8,22]. These consequent changes in plant species composition may lead to poor rangeland conditions and less fodder for herbivores (especially grazers) [23–25]. Rangelands in a poor condition are often characterised by open and bare patches with degraded soil. Such patches are more exposed to extreme temperatures, wind and water, leading to erosion and decreased soil nutrient and moisture content [24,26,27]. Open, bare patches are also more readily colonised by ephemerals, changing the type of patches occurring in the habitat [8,26,28]. Consequently, the abundance of more palatable perennial grass species decreases and is replaced by less palatable perennial and annual species which are often toxic to grazers [12,23,25,29]. The latter are weak competitors due to their shallower root systems. Woody species mostly have a deeper and stronger tap-root system that can absorb deeper soil water, leading to an increase in the density of the woody component, especially in the absence of competition due to a loss in the herbaceous layer [22,28,30,31]. This phenomenon is most prominent under drought conditions in the sandy soils of the Molopo region in the North West Province of South Africa where this study was conducted [32].

Shifts in vegetation composition can affect the ecosystem functions provided by globally important savanna landscapes [10] and, in particular, impact the provision of ecosystem services to land users who attempt to implement sustainable land management practices in savanna rangelands [33,34]. Consequently, bush encroachment poses both ecological and economic challenges to savannas globally [13,35–38].

In an attempt to control bush encroachment and to restore rangelands to a more stable and productive state, rangeland managers resort to various methods but, seemingly, favour soil arboricides that are applied either manually (by hand) or aerially (by aeroplane) [32,39–41]. Rangeland managers who prefer not to use arboricides, due to their non-selective nature, usually resort to manual methods of bush clearing such as burning stems, cutting stumps or bulldozing woody species [19,32,40,41]. Irrespective of the method employed to curb bush encroachment, regular follow-up treatments following on bush control are of great significance since several studies [8,16,39] reported that bush thickening can re-occur within a period of approximately three to five years, with some woody species being even more aggressive, resulting in denser, impenetrable thickets as found by the owner (Mr. Johan Olivier, Farm: Jones, PO Box 1237, Kuruman, South Africa, 8460, E-mail: jjolovier@xsinet.co.za) of the farm Jones in the Kuruman district. Undoubtedly, the successful curbing/control of bush encroachment is dependent on a long-term commitment by land owners and rangeland managers [8,16,32].

To contribute to the application of sustainable management practices aimed at enhancing the economic profitability, conservation and restoration of degraded semi-arid savanna rangelands, this study set out to gain a better understanding of the effects bush encroachment and various control methods have on landscape functionality and recovery as well as soil parameters. Bush-control measures ranging from stem burning and hand control to aerial spraying have been implemented in the Molopo region from as early as 1982, with the last treatment applied to the study area as recently as 2012.

Given the study area's protracted history with regards to controlling bush encroachment, the landscape function analysis (LFA) monitoring procedure was used to describe and assesses the level at which a landscape is functioning as a biophysical system by comparing bush-controlled sites with still bush-encroached sites as the reference [42].

## 2. Materials and Methods

### 2.1. Study Area

This study was conducted in the Molopo region (an area commonly referred to as 'the Molopo'), a typical Savanna area that spans the North West and Northern Cape Provinces of South Africa (Figures 1 and 2). The region borders on Botswana, with the Molopo River forming the northern border [43,44].

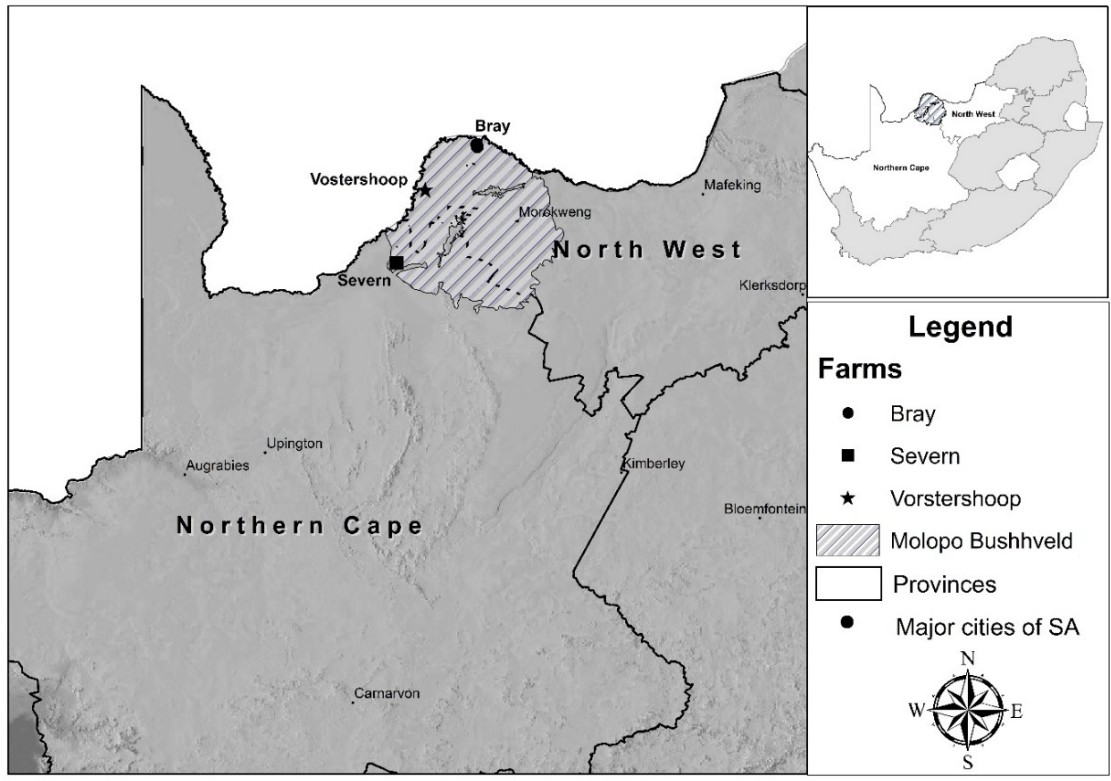

**Figure 1.** Extent of the Molopo Bushveld region in the North West and Northern Cape Provinces of South Africa with the location of the study areas (See Appendix A for details).

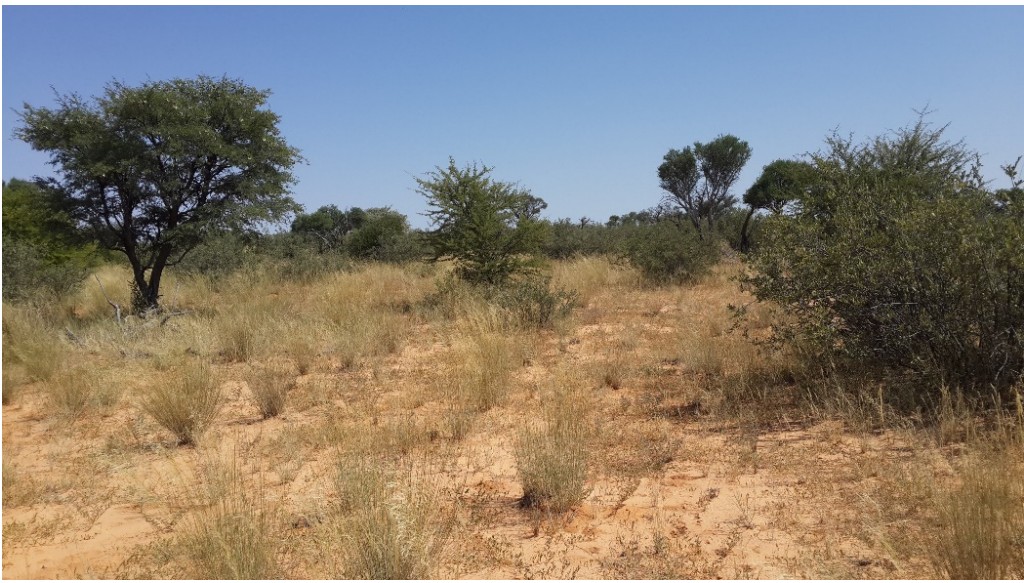

**Figure 2.** Typical Savanna area of the Molopo region in the semi-arid parts of the Northern Cape and North West Provinces in Southern Africa, where the study was conducted.

The Molopo receives erratic summer rainfall ranging between 250 and 400 mm per annum [45–49] interspersed with frequent dry spells leading to droughts [50,51]. One such extreme drought occurred from October 2014 until April 2016, coinciding with this study's vegetation sampling in February and March of 2015 and 2016. The two weather stations closest to the study sites (i.e., Bray and Severn) recorded the long-term average rainfall from October to February for the period 1986 to 2015 as 241 and 189 mm/a respectively (Figure 3). As is evident in Figure 4, these values decreased greatly for the four months preceding the vegetation surveys in 2015 and 2016: Bray recorded only 65 mm of rainfall from October 2014 to February 2015, whereas no rainfall was recorded between October 2015 and February 2016. Only 134 mm was recorded for the Severn area from October 2014 to February 2015 and as little as 51 mm from October 2015 to February 2016. Even the South African Weather Service considered this drought event as the most severe since 1950.

Characterised by flat to sinuate sandy plains, the topography of the study area (largely located on the Molopo Farm Complex) varies between 1000 and 1300 m above sea level [48,52] and is comprised of layered ultramafic-mafic intrusions of the Kaapvaal Craton [53] with Kalahari sands of the Cenozoic Kalahari Group as well as Karoo strata occurring mostly in the northern parts of the North-West Province [45,54,55]. These sandy plains are underlain by the Cordonian Formation with red, deep aeolian sands. Furthermore, silcrete, calcrete and ferricrete characterise the soil surface [48,56]. Kalahari soils are well-drained with a high base status [48] as well as a high infiltration rate and a low organic matter and nutrient content with limited water run-off [46,52]. The deep sandy soils (>1.2 m) [48] are characterised by a coarse soil texture together with low clay content. Consequently, water, air and plant roots can easily penetrate these soils, resulting in a decreased capacity to retain water and nutrients [57,58].

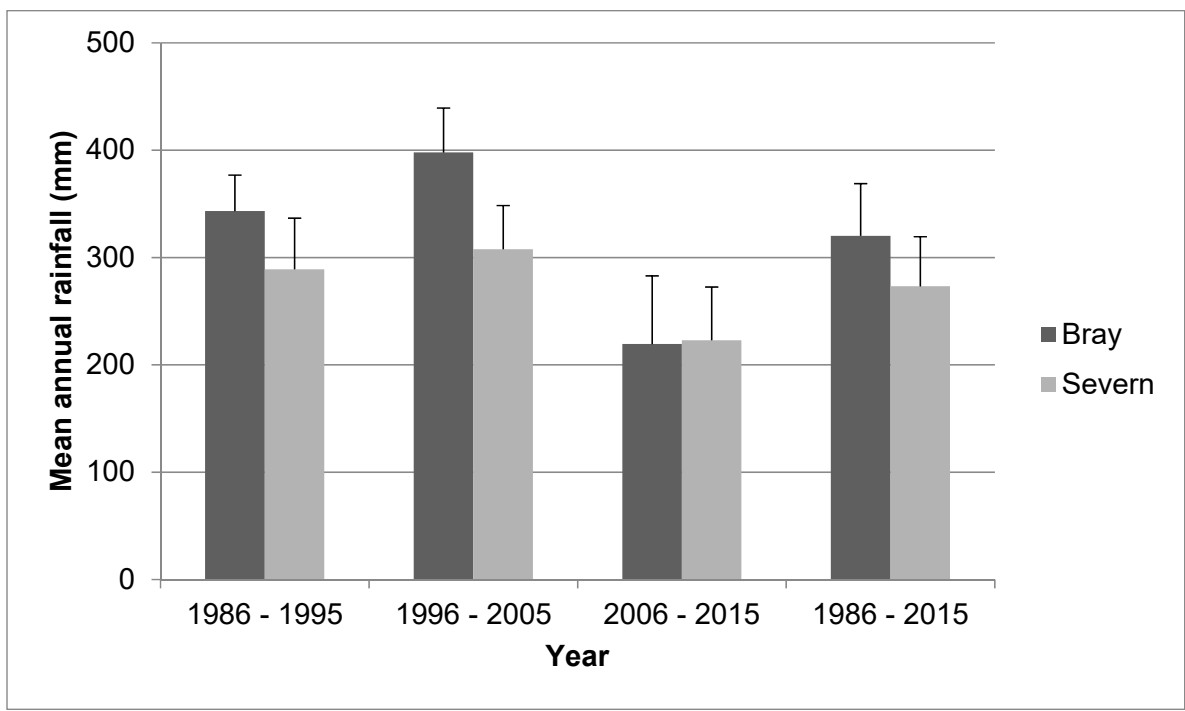

**Figure 3.** Mean annual rainfall (±coefficient of variation) recorded over three decades by the Bray and Severn weather stations reflected alongside the long-term mean annual rainfall for the period 1986 to 2015.

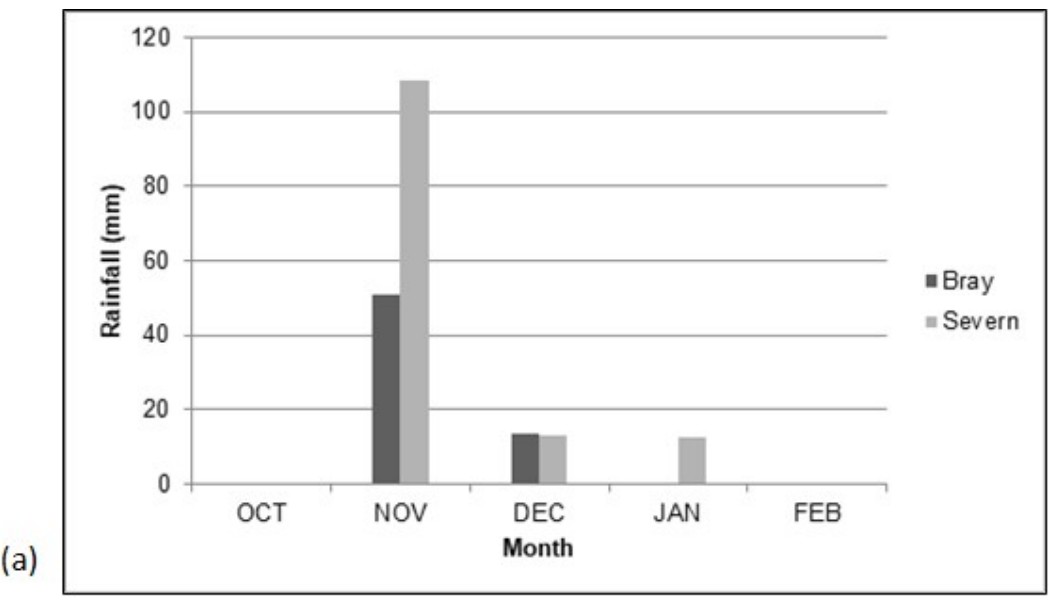

(a)

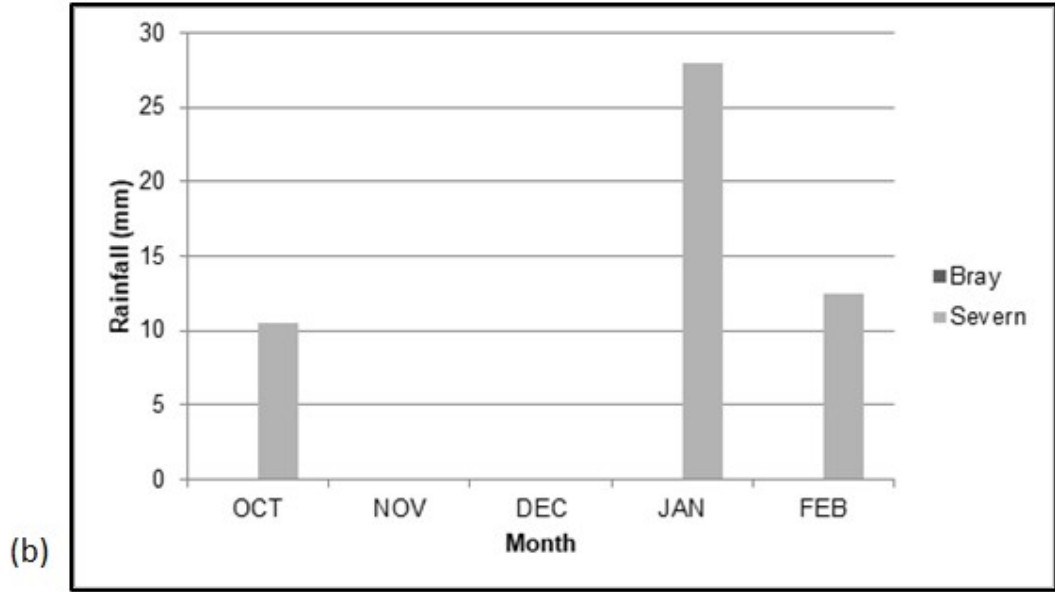

(b)

**Figure 4.** Rainfall in the months preceding the (**a**) 2015 field surveys (end-February 2015 till the beginning of March 2015) and the (**b**) 2016 field surveys (from the beginning till the end of March 2016). Note that Bray recorded no rainfall during these periods.

The Molopo Bushveld vegetation type (SVk 11) of the Eastern Kalahari Bushveld dominates the study area [48]. The tree layer consists mostly of scattered individuals of species such as *Boscia albitrunca, Terminalia sericea, Vachellia erioloba* and *V. luederitzii* var. *luederitzii.* Many degraded areas are characterised by a shrub layer comprised of *Grewia flava, Lycium hirsutum, V. haematoxylon, Senegalia mellifera* subsp. *detinens* and *V. hebeclada* subsp. *hebeclada* [44]. The herbaceous layer is well developed and dominated by perennial grasses such as *Aristida stipitata, A. meridionalis, Eragrostis lehmanniana, Schmidtia pappophoroides* and *Stipagrostis uniplumis* [44,56,59] as well as the annuals *Enneapogon desvauxii* and *Schmidtia kalahariensis* [44,56,59,60].

The Molopo region is classified as moderately degraded based on the combined degradation index (CDI) [7]. Bush encroachment has escalated in the region with serious implications for commercial and subsistence farmers [61]. As mentioned, landowners in the Molopo use arboricides to control bush encroachment in attempts to increase fodder

production for livestock and game [8,16,41,43,60,62,63]. The most effective arboricide preferred by landowners is Molopo CC granules (with tebuthiuron as the active ingredient) that can be applied selectively by hand or non-selectively by aeroplane [44]. Aerial bush control is a faster method than control by hand since larger areas can be treated in a shorter time period. However, it requires more financial inputs especially when dense shrub stands need to be treated [41,60].

Landowners, especially cattle farmers, use rotational grazing regimes whereby rangelands are divided into various smaller paddocks [44] and stocking rates of between 10 and 12 large stock units per hectare are applied [44,60,64]. When appropriately managed strategies with sufficient resting periods between paddocks are applied, rangelands are better protected against bush encroachment over the long term [64]. Resting periods usually occur after good rainfall events to promote recovery of palatable perennial grasses [23]. During drought years, though, pastures remain exposed to pressure from livestock (e.g., trampling, severe overgrazing) and are not rested despite the retraction of livestock from a paddock.

*2.2. Experimental Design and Sampling*

Prior to vegetation surveys, qualitative data such as paddock size, grazing pressure and stocking rate regimes, the condition of the grass layer and the soil condition of the study sites were documented. Additionally, general information (i.e., GPS coordinates, site number, survey date, dominant plant species, land use, disturbance, soil type, topography) was also documented. The vegetation and soil sampling methods followed the LFA monitoring approach [42].

Landscape Function Analysis (LFA) is a field procedure using readily observed indicators to assess and monitor soil properties as they affect plant growth at the hillslope and patch scale. The indicators represent physical, chemical and biological soil properties in terms of dynamic processes rather than static properties such as colour. There are 11 indicators which together summarise soil stability or resistance to erosion, water infiltration rate and storage and nutrient cycling processes. The approach is comprised of a conceptual framework, a field data collection protocol and a reporting framework, enabling observers to place the functional state of a disturbed study site in context with undisturbed reference sites 42]. In our study we identified and mapped "patches" where mobile resources tend to be accumulated and "inter-patches" where resources tend to be mobilised and transported downslope. The LFA indicators enabled us to quantify differences in soil productive potential between patches and inter-patches. We used these differences to compare and contrast bush-thickened and bush-controlled sites.

Accordingly, the LFA monitoring procedure was applied to five bush-control treatments: (1) selective chemical control by hand (hand controlled—HC); (2) selective chemical control by hand with re-application (selective re-application by hand—2HC); (3) non-selective aerial chemical control (aeroplane-controlled—AC); (4) selective control by stem burning with re-application (stem burning—SB); and (5) sustainably managed (SM). The control site remained bush-encroached (thickened) (BT). The respective sampling sites with various treatments were located on three commercial cattle farms and one game ranch in the Molopo region (Figure 1; Table A1; Appendices A and B). Sites were selected in collaboration with the landowners. The environmental conditions (rainfall, soil type, etc.) at all sites were similar. Sampling sites representing two transects were placed approximately 500 m apart in treated areas and representative landscapes. No paddocks occurred on the game ranch, whereas the commercially managed cattle farms followed a four, six and eight rotational management system (Table A1). As mentioned, sampling was conducted over two seasons in 2015 and 2016. Two LFAs were conducted on 50 m transects at each site (i.e., 41 sites in total across various treatments; Table A1). The two transects were laid out at random, approximately 30–40 m apart within each treatment. LFA data were collected according to three scales: (i) an overall landscape description; (ii) a landscape organisation scale; and (iii) a fine scale using a soil surface assessment of 11 indicators.

Following Tongway and Hindley's manual, each LFA commenced with a landscape organisation analysis (LO) that entailed dividing the 'landscape' on the transect into various patch zones which represented different obstructions that could trap mobile resources or allow resources to move freely (i.e., resource accumulating zones, inter-patch zones). Transects were then laid out in the direction of water flow (i.e., from higher to lower) to describe effectively how resources could be trapped. These transects are commonly known as gradsects, i.e., gradient-orientated transects, and their size (width and length), nature and location were identified and documented on a prescribed data sheet as per the manual [42].

Patch types included inter-patches (Ips), grass patches (GPs), grass litter patches (GLPs), litter patches (LPs), shrub litter patches (SLPs) and grass shrub patches (GSPs), with the addition of shrub patches (SPs) that have been identified in 2015 only (See Appendix B for a full description of patch types). Soil surface assessments (SSA) were carried out on each gradsect in 3–5 query zones representing the identified patch types. In the course of the SSAs, eleven indicators as identified by the LFA methodology [42] were assessed in the gradsect of each patch type. (These indicators were then randomly combined with the aid of a prescribed spreadsheet to synthesise three landscape function parameters, i.e., stability, infiltration and nutrient cycling.

Here it ought to be noted that the drought the Molopo region experienced in the summer of 2015/2016 had a severe effect on the general length and width of all identified patches as well as the frequency of resource accumulation in the respective identified patches [65].

*2.3. Soil Analysis*

Following the completion of the landscape organisation, the two most dominant patch types were identified along each transect. At each of these patches, three soil samples were collected up to a depth of 5 cm. The three samples were then combined to form a composite sample. Care was taken not to include any litter in the samples. Although soil samples were collected during both sampling seasons, only samples collected in 2015 were analysed for chemical properties.

Composite samples were analysed for pH (KCL), macro- and micronutrient status (Ca, Mg K and Na), available phosphorus (P-Bray 1) and soil carbon (%C): pH (KCL) values were determined using a 1:2.5 extract solution method [66], macro- and micronutrient content by using a 1:2 extract method and available phosphorus by using a P-Bray 1 solution [67], while carbon content was determined by loss on ignition [68].

Note that all analyses were performed by Eco-Analytica, a laboratory dedicated to natural and agricultural sciences housed on North-West University's Potchefstroom campus, in accordance with the methods and procedures prescribed by the Non-affiliated Soil Analysis Work Committee (1990).

*2.4. Statistical Analysis*

The three main indices derived from the LFA monitoring procedure (i.e., stability, infiltration and nutrient cycling) as well as the landscape organisation index were used to compare and analyse the effects bush encroachment and the respective control methods had on the functional and structural components of the study sites in the Molopo savanna landscape. A Wilcoxon test was conducted to analyse the landscape organisation index and three LFA indices to determine whether their distribution was structured (parametric) or unstructured (non-parametric). Mixed-model analyses were performed in Statistica version 14 [69] to determine whether there were any significant differences between these indices across the respective treatments. Multivariate analyses, e.g., a principal component analysis (PCA), were performed with the aid of Canoco version 5 [70] to determine whether associations could be found between the LFA indices, landscape organisation index and bush density of woody species as well as to determine which patches were associated with the respective soil properties.

## 3. Results

### 3.1. LFA Indices

Results from the LFAs conducted across treatments are summarised in Table 1. Considering the three LFA indices as percentages in the two sampling years, index values across bush-controlled treatments, as well as for the sustainably managed and bush-thickened sites, remained below 50%. Of all three indices, the stability index scored the highest, followed by infiltration and nutrient cycling. Therefore, stability is considered the most important factor with regards to the overall functioning of this landscape, contributing approximately between 40 and 50% across bush-controlled (Figure 5) and bush-encroached (Figure 6) sites.

Mixed-model analysis revealed non-significant differences ($p > 0.05$) for bush-controlled treatments (Figure 5) and bush-encroached sites (Figure 6) with regards to all three LFA indices. Comparing 2016 to 2015, LOI improved slightly at the aeroplane- and sustainably managed sites, while landscape organisation index (LOI) values for stem-burnt and hand-controlled sites declined (Table 1). Surprisingly, LOI remained the same for bush-encroached sites.

**Table 1.** Results of the landscape function analysis (LFA), indicating the cumulative scores for the LFA indices and the landscape organisational index (LOI) for the 2015 and 2016 surveys across treatments.

| Treatment | LOI | | Stability | | Infiltration | | Nutrient Cycling | |
|---|---|---|---|---|---|---|---|---|
| | 2015 | 2016 | 2015 | 2016 | 2015 | 2016 | 2015 | 2016 |
| Aeroplane | 0.21 | 0.32 | 43.00 | 42.50 | 34.50 | 36.50 | 13.00 | 14.00 |
| Stem burning | 0.51 | 0.32 | 43.00 | 41.00 | 35.00 | 37.00 | 14.00 | 14.00 |
| Hand control | 0.26 | 0.13 | 45.33 | 43.00 | 36.00 | 36.67 | 14.67 | 15.00 |
| Bush thickened | 0.31 | 0.31 | 44.25 | 44.75 | 35.75 | 36.75 | 14.00 | 15.50 |
| Sustainably managed | 0.28 | 0.29 | 44.00 | 44.00 | 35.00 | 36.00 | 14.00 | 14.00 |

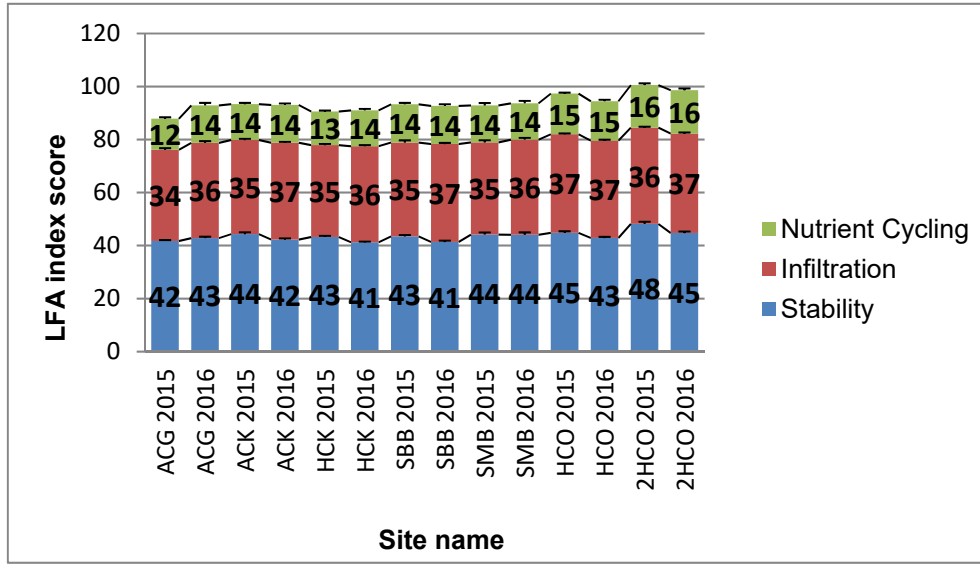

**Figure 5.** Average scores of the three landscape function analysis (LFA) indices across bush-controlled sites in 2015 and 2016. The standard error is indicated by error bars. No significant differences were found. AC—aeroplane-controlled; SB—stem burning; HC—hand controlled; BT—bush encroached/thickened; SM—sustainably managed.

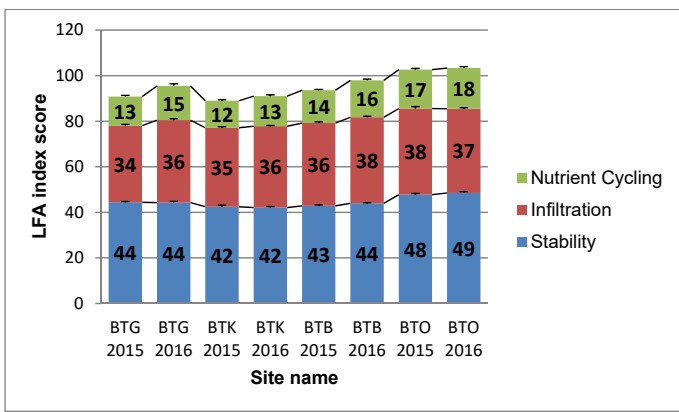

**Figure 6.** Average scores of the three landscape function analysis (LFA) indices for bush-encroached sites for both survey years (2015 and 2016). The standard error is indicated by error bars. No significant differences were revealed between LFA index scores.

Stability over time decreased on the aeroplane-controlled, stem-burnt and hand-controlled sites and increased in bush-encroached sites, whereas sustainably managed sites had the same stability. Interestingly, the stability of bush-encroached sites did not differ from those of the bush-controlled sites despite larger inter-patches recorded in most bush-encroached sites, whilst bush-controlled and bush-encroached sites reflected similar infiltration rates (Table 1).

Nutrient cycling for stem-burnt and sustainably managed sites did not improve from 2015 (Table 1). In 2016, nutrient cycling values showed slight improvements for the aerially and hand-controlled sites as well as for bush-encroached sites, with the latter being the highest due to the high occurrence of cryptogams.

*3.2. Association of Bush-Controlled and Bush-Encroached Sites Based on LFA Index Scores*

The first redundancy analysis axis (RDA) explained 62.1% of the total variation while the second axis explained 37.9% (Figure 7). A clear nutrient cycling, LOI and bush density (TE/ha) gradient was revealed along the first axis with clear clustering between treatments. The PCA ordination clustered aeroplane-controlled sites with their low (i) nutrient cycling, (ii) LOI and (iii) bush densities towards the positive plane. These sites were also positively correlated with high stability and infiltration. Bush-encroached sites were grouped towards the negative plane due to the higher (i) nutrient cycling, (ii) LOI and (iii) woody density. Between aeroplane-controlled and bush-encroached sites, hand-controlled sites clustered, indicating intermediate levels of nutrient cycling, LOI and bush density.

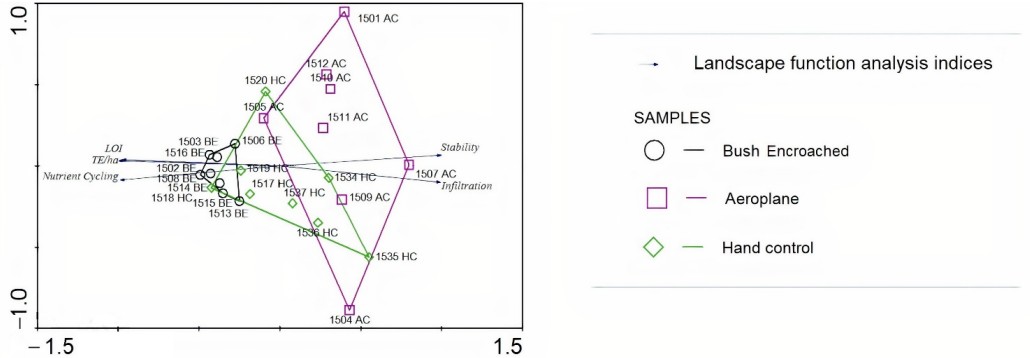

**Figure 7.** Redundancy analysis (RDA) biplot indicating association of bush-thickened and bush-controlled sites based on landscape function analysis (LFA) index scores for the year 2015. Bush density is indicated by tree equivalent per hectare (TE/ha).

### 3.3. Patch Type Associations with Soil Chemical Properties

Inter-patches, grass patches, grass litter patches and shrub litter patches occurred most frequently (i.e., three times or more) across the various treatments and were, therefore, considered to be the dominant patch types. Bush-encroached sites were dominated by inter-patches and shrub litter patches which occurred 19 and 15 times, respectively. The bush-controlled sites (SM, SB, AC, HC and 2HC) were dominated by inter-patches and grass litter patches which were encountered 31 and 23 times, respectively, across the treatments. Due to the drought conditions, all sites were still heavily utilized by herbivores, causing disturbances by hoof actions at all sites with very little soil crusts.

The first two RDA axes explained 71.5% (0.500 + 0.215) of the variability in patch type data (Figure 8). The first axis mainly correlated with HC, BT, SM and SB towards the left and AC to the right of the ordination plane, which explained the largest proportion of variance (i.e., 50%; Figure 5).

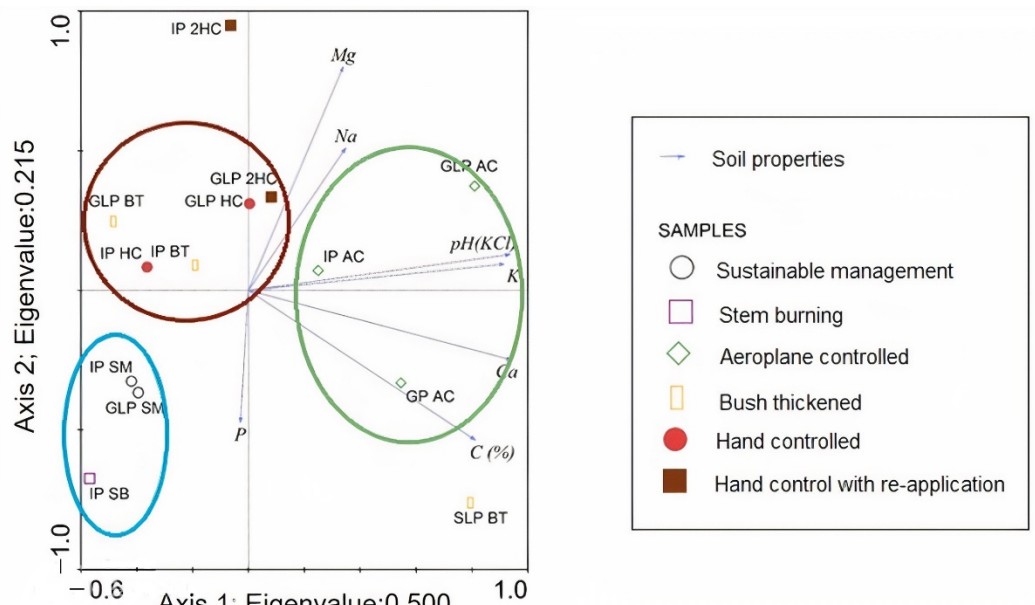

**Figure 8.** Redundancy Analysis (RDA) biplot indicating associations of dominant patches from bush thickened/encroached (BT) and bush-controlled sites.

Three distinct groupings could be identified from the RDA (Figure 8). Firstly, inter-patches (IP) and grass litter patches (GLP) in sustainably managed (SM) and stem-burned (SB) sites differed from other treatments, especially with respect to high phosphorous (P) levels (Figure 8). However, inter-patches and grass litter patches of sustainably managed and stem-burned sites did not correlate strongly with any of the other chemical properties (Figure 8). Soils from the inter-patches and grass litter patches in the HC and 2HC treatments correlated strongly with one another and had higher levels of magnesium (Mg), while sodium (Na) was only elevated in the 2HC treatments (Figure 8). The remaining patches found in this grouping (i.e., grass litter patches of the bush-encroached sites and inter-patches of the hand-controlled sites) were not similarly associated with either Mg or Na and, furthermore, revealed the weakest correlation with calcium (Ca), potassium (K), carbon I and soil pH (Figure 8), which was comparably lower (4.8—Table 2). The soil of the AC patch types (inter-patches, grass patches and grass litter patches) illustrated the strongest association with high levels of Ca, Mg, K and C (Figure 5) and revealed the highest pH values across all dominant patch types (5.3 to 5.6—Table 2). The highest levels of Mg, K and Na were associated with the grass litter patches of the AC treatment (Figure 8), whilst grass patches revealed the strongest association with Ca and C (Figure 8). The shrub litter patches of the bush-encroached site, in particular, were very strongly associated with

the Ca and C content (Figure 8) and, in addition, correlated with a high pH value (5.3) similar to the inter-patches and grass patches of the aeroplane-controlled sites (Table 2).

**Table 2.** Summary of the nutrient status from the soil samples collected at the various treatments in the four dominant patch types (inter-patch (IP), grass patch (GP), shrub litter patch (SLP) and grass litter patch (GLP)) during the 2015 survey.

| Treatments | Patch Type | Ca | Mg | K | Na | P | pH (KCl) | C (%) |
|---|---|---|---|---|---|---|---|---|
| | | | | (mg/kg) | | | | |
| **Aeroplane-controlled** | IP | 209.0 | 60.0 | 82.1 | 3.0 | 8.7 | 5.3 | 0.3 |
| | GLP | 256.8 | 96.6 | 111.1 | 5.4 | 10.3 | 5.6 | 0.4 |
| | GP | 299.9 | 40.4 | 98.3 | 2.1 | 7.7 | 5.3 | 0.4 |
| **Hand controlled** | IP | 124.0 | 65 | 48.5 | 0.5 | 8 | 4.8 | 0.3 |
| | GLP | 148.3 | 81.5 | 64.7 | 0.8 | 7.8 | 5.2 | 0.3 |
| **Hand controlled with re-application** | IP | 147 | 82.7 | 66.8 | 6.3 | 7.5 | 4.8 | 0.3 |
| | GLP | 190.5 | 91.3 | 64.2 | 2.2 | 8.9 | 5.1 | 0.3 |
| **Sustainably managed** | IP | 119 | 47.7 | 52.2 | 2.8 | 10.9 | 4.5 | 0.3 |
| | GLP | 113.4 | 49.1 | 57.3 | 1.8 | 10.2 | 4.5 | 0.3 |
| **Stem burning** | IP | 114 | 21 | 40.8 | 1.2 | 9.5 | 4.5 | 0.3 |
| **Bush encroached** | IP | 167.8 | 47.4 | 48.4 | 4.7 | 9.3 | 5 | 0.3 |
| | GLP | 108.0 | 69.2 | 37.7 | 1.2 | 8.4 | 4.8 | 0.3 |
| | SLP | 350.7 | 53.0 | 78.9 | 2.2 | 9.6 | 5.6 | 0.5 |

*3.4. Soil Chemical Properties across Treatments*

Of the various treatments, grass litter patches at the aeroplane-controlled (AC) sites had the highest Mg (96.6 mg/kg/), K (111.1 mg/kg) and pH levels (5.6) overall of all the patch types (Table 2) and, in addition, had the second highest values for Na (5.4 mg/kg), P (10.3 mg/kg) and C (0.4%) and the third highest value for Ca (256.8 mg/kg).

According to the FSSA (2007), the Mg content of agricultural soil should be 50–300 mg/kg, with 50 mg/kg being very low. Five of the sampled patches (inter-patches of the stem-burned site, inter-patches and grass litter patches of the sustainably managed site, and grass patches and inter-patches of the bush-thickened site) scored below 50 mg/kg with the lowest content recorded in the inter-patch of the stem-burned site, namely 21 mg/kg (Table 2). The patch with the highest Mg content was the grass litter patch of the aeroplane-controlled site (96 mg/kg), which is low compared to the FSSA (2007) standards. Although Ca levels were high for all treatments (Table 2), a trend similar to that of Mg levels, only four patches (inter-patch, grass patch and grass litter patch of the aeroplane-controlled site and shrub litter patch of the bush-thickened site) revealed higher levels than the lowest acceptable range of FSSA (2007) standards. K levels conformed better to the FSSA (2007) standards of 40–250 mg/kg. Grass litter patches of the aeroplane-controlled site revealed a K value of 111.1 mg/kg which was the highest recorded for all sites and treatments, followed by the other two patches from the same site (i.e., grass patch and inter-patch) at 98.3 and 82.1 mg/kg, respectively (Table 2). Only one of the patches (grass litter patch of the bush-thickened site) had K levels below 40 mg/kg (Table 2).

An interesting result was found in the patches of the sustainably managed (SM) and stem burned (SB) sites. Grass densities were high at these sites, leading to high LOI values. Both these sites, however, had the lowest nutrient concentration values, except for P (Table 2). The inter-patches and grass litter patches of the SM sites scored the first and third highest P levels of all patches at the various sites, respectively. The pH levels recorded at these sites were also the lowest of all the patches at the various sites, with all three patches recording a pH of 4.5 (Table 2).

Bush-encroached sites only scored the highest in three of the seven parameters measured in the soil analysis, namely Ca content, pH and C (%). This can be attributed to the increased woody litter present in these sites. The Ca concentrations of the soil at the shrub litter patches (350.7 mg/kg) were much higher than that of other patches.

Grass patches and grass litter patches from the aeroplane-controlled sites had Ca concentrations of 299.9 and 256.8 mg/kg respectively.

The Ca concentrations of soils at the inter-patches and grass litter patches from the same bush-thickened sites were less than half that of the shrub litter patches, namely 167.8 and 108 mg/kg, respectively (Table 2). The C (%) of the soil at the shrub litter patch samples followed the same trend as that of the Ca concentrations, namely that the second and third highest concentrations were found in the soils of the grass patches and grass litter patches of the aeroplane-controlled sites (Table 2).

Soil from the hand-controlled (HC) sites had surprisingly low nutrient levels.

Mg in the soil of the grass litter patches was the only soil nutrient with a reasonably high concentration, i.e., 81.5 mg/kg (fourth highest). The concentrations of other nutrients in the grass litter patches and inter-patches were third or fourth lowest, which was an unexpected result as the HC sites on average scored the second highest (after hand-controlled sites with re-application) in most of the LFA indices over both survey years.

Na levels across the soil samples were low, with most levels being around 2 mg/kg. The only patches with moderately high soil nutrient levels were inter-patches of the hand-controlled sites with re-application, grass litter patches of the aeroplane-controlled sites and inter-patches of the bush-thickened sites with 6.3, 5.4 and 4.7 mg/kg, respectively (Table 2).

## 4. Discussion

### 4.1. Landscape Function of Bush-Controlled and Bush-Encroached Sites

Stability is an important factor when describing the rate and extent of erosion. Since no to very few rainfall events occurred before the sampling period, and given that the study sites were located on flat surfaces, erosion by especially water was low, which contributed to the higher stability values. With regards to infiltration rates, it was unexpected that bush-controlled and bush-encroached sites yielded similar infiltration results, especially when considering that more inter-patches were found at bush-encroached sites than on the bush-controlled sites. In general, increases in inter-patches would result in higher rates of water and nutrient flow from the system. Hence low infiltration rates are caused due to a lack of patches that can prevent the flow of resources. Thus, irrespective of treatment, infiltration values improved from 2015. The low nutrient cycling values revealed in this study correspond with the findings of [46,52,71,72] who all found nutrient cycling of the sandy soils in the Molopo to be low.

In general, landscape functioning of the bush-encroached and bush-controlled sites was similar when using the LFA methodology. This study found that the bush-encroached areas in the Molopo region were more functional with regards to soil stability and soil nutrients. This can be attributed to the higher densities of the woody canopies in the bush-encroached sites that have extensive root systems stabilising the loose sandy soils and preventing soil erosion [36,73–77]. Root systems also increase soil nutrient cycling by extracting minerals from deep soil layers in the soil profile and depositing those minerals close to the surface [77]. Canopies formed by woody species within bush-encroached sites are also considered to facilitate the establishment of cryptogams beneath those canopies, which further increases nitrogen fixation and soil stability as these form a biological soil crust that is not grazed [78,79]. The cryptogams forming under canopies in the bush-thickened sites can produce extracellular polysaccharides which increase soil nutrients by fixing nitrogen and carbon that form part of biological soil crusts [78,80]. Both [63,78] found that cryptogams are also responsible for increased soil nutrients that are predominantly found in bush-encroached sites beneath *Grewia flava* and *Senegalia mellifera* shrubs. The higher LOI scores found in some of the bush-encroached sites can be ascribed to the large shrub litter patches and shrub patches formed by *S. mellifera*. This relationship between *S. mellifera*

shrubs and cryptogams can be one of the main reasons why bush-encroached sites had higher stability and nutrient cycling scores, resulting in higher landscape functionality. Furthermore, with increased canopy cover, rainfall is intercepted implying that water flow is limited to the soil beneath trees, resulting in lower soil erosion and soil movement [81]. Additionally, more shrub cover limits wind erosion at sites that consist of bare soil [74]. Infiltration rates under woody canopies have also been found to be higher compared to inter-canopy areas and could, therefore, influence available soil moisture [82].

Bush-control methods should aim to be more selective as such methods have been found to be least harmful to the environment [32]. Furthermore, the average woody density at the hand-controlled sites was almost double the aeroplane-controlled sites and half of the bush-encroached sites. Large individuals of *Vachellia erioloba* and *Boscia albitrunca* were also more represented within the selective chemically controlled sites. These species provide fodder as well as shade for livestock and game. Larger trees also prevent the re-establishment of *S. mellifera* seedlings through competition [12]. Furthermore, *G. flava* shrubs dominated in the manually controlled sites. Additionally, favourable woody species (i.e., *B. albitrunca*, *G. flava* and *V. erioloba*) also increase the recovery time of rangelands by creating fertile islands with elevated soil nutrient levels [77,83], thereby promoting herbaceous production over the long term.

LOI scores associated with bush-encroached sites were mostly related to the size difference and frequency of dominant patches (i.e., nutrient-accumulating patches) present at these sites. These were mostly shrub litter patches in combination with high cryptogram densities beneath *G. flava* and *S. mellifera* shrubs. At aeroplane- and hand-controlled sites, more grass litter patches and grass shrub patches occurred with increased nutrient accumulation, mainly due to the robustness of the shrub litter in the environment. This study revealed that landscape functionality of bush-controlled sites does not differ from bush-encroached/-thickened sites. These results are supported by various studies [36,75–77,84,85] where the researchers found that the effects of bush encroachment on an ecosystem depend on end-use. Livestock farmers believe that bush-encroached sites are unfavourable since grass production and density are reduced [8,44,60]. In contrast, others consider bush encroachment as a natural occurrence that increases ecosystem services (i.e., biodiversity, carbon sequestration, soil fertility and hydrology), but only if bush encroachment does not increase beyond a certain degradation threshold [76,77,85].

Further research is, however, required over longer periods of time and across different areas where the LFA transects are laid out to validate the above-mentioned findings. More LFA surveys are, therefore, needed to increase the amount of data available for statistical analysis. Results from LFAs conducted in good or average rainfall years will further our understanding regarding landscape functionality between bush-thickened and bush-controlled sites.

### 4.2. Soil Properties across Treatements and Patch Types

The high nutrient levels documented for the grass litter patches found in the AC sites can possibly be a result of the dominant grass species, *Stipagrostis uniplumis*, found throughout these sites, as well as the Ca fraction that occurs in the arboricide granules. *Stipagrostis uniplumis* is a large perennial, tufted grass that can produce large volumes of plant material/litter if exposed to favourable conditions. Biological soil crusts or cryptogams also formed around the base of this grass species in the larger grass litter patches, especially where more shade occurred. Both these occurrences can lead to elevated nutrient levels in the soil surrounding the grass litter patches [78,86,87]. This could also be the reason for the elevated nutrient levels of the grass patches sampled at the same sites.

The patch with the highest Mg content was the grass litter patch of the aeroplane-controlled site (96 mg/kg), which is low compared to the FSSA (2007) standards. Mg deficiency in plants can cause roots and shoots to become smaller and shorter with necrotic spots forming on the leaves. These symptoms are a result of impaired C metabolism and a decrease in the overall chlorophyll and C fixation rates [88,89]. Mg deficiencies in the

soil can be ascribed to a number of factors, such as high Na, K or Ca levels or leaching, especially in sandy soils [89,90]. Na, K and Ca are strong competitors and can replace Mg, leading to decreased availability of this element [89]. In the Molopo, Mg leaching occurs due to the soil's high infiltration rates [46,52] and the high mobility of Mg in soil [90], possibly explaining the low levels of Mg found in the soil of the study sites.

Ca plays a crucial role in the strengthening of cell walls and the protection of the plant against diseases and, most importantly, heat stress (especially in the dry Molopo region) [90,91]. Ca deficiency in plants can lead to the dieback or scorching of young leaves due to reduced transpiration rates [91], and Ca shortages can be caused by acidic soils or high levels of other positively charged ions, including Mg, Na and K [90,91]. However, in this case, the high levels of cations cannot be the reason for the Ca levels of the samples being low. Furthermore, Ca is not very mobile in soil, limiting the effect leaching might have on its availability for the plants [90,92]. One other remaining factor that could possibly explain the limited availability of Ca is the pH of the soil and the possible antagonistic behaviour between macro- and microelements. Woody and grass litter as well as the partial amelioration effect provided by chemical control were considered to contribute to Ca concentrations associated with grass patches and grass litter patches at aeroplane-controlled sites. Low soil nutrient levels revealed for HC sites were related to the selective and patchy distribution of the hand application process, which causes a more unequal distribution of nutrients that can be more easily overlooked by low frequency sampling of the soil.

P is a primary plant macronutrient needed but is not as readily absorbed as N or K. It plays a key part in fundamental plant processes, including photosynthesis, N fixation and maturation [93]. P deficiencies can be difficult to identify, but more mature plants tend to have darkened leaves, with severe deficiencies leading to yellowing and senescence of leaves [93]. In the soil profile, P is mostly available as inorganic phosphate ions (e.g., $HPO_4^{2-}$ and $H_2PO_4^{2-}$) with the former being more readily available in acidic soils [93] similar to those found in the Molopo region. Additionally, the high P content associated with sustainably managed (SM) and stem burned (SB) sites is possibly a result of nutrient cycling. The paddocks used in the rotational grazing system contribute to higher densities of grazing animals on these soils leading to higher concentrations of manure being recycled into the soil, which is one of the second major sources of P.

The high amounts of litter and the higher shrub densities created the perception that the overall nutrient levels of the hand-controlled sites would be reasonably high. The disturbance caused by the high grazing pressure experienced at the HC sites in 2015 probably influenced the nutrient levels in the top 5 cm of the soil profile negatively. Both HC sites were situated on cattle farms and grazed just before or during the 2015 surveys. The 2HC sites, also located on cattle farms, were grazed prior to sampling. The difference between the 2HC and HC sites was the amount of litter generated by woody species. The contractors who controlled the woody species in the pasture did not apply selective control, meaning that all woody species, not only problem species, were chemically controlled, causing further bush encroachment. Dead tree material or litter of the controlled woody species were left behind by the contractors, which could have led to the overall higher nutrient levels in the pastures, thereby contributing to the higher LFA indices, especially the C (%) values in the soils of the grass patches, grass litter patches and shrub litter patches.

K is one of the most important plant nutrients after N [91], especially because K cations can be replaced by other cations if the latter are present in large quantities [90]. This can have detrimental effects on plants since K helps to regulate $CO_2$ uptake by playing a role in the opening and closing of stomata. Furthermore, K helps with the osmoregulation of plants to compensate for water loss through the stomata [91,92]. K deficiency can, therefore, cause chlorosis or scorching of the leaves or stunted growth. The most detrimental effect of K deficiency on plants in the Molopo region is reduced resistance to fluctuations in temperature and drought [91,92].

According to the FSSA (2007) guidelines, Na levels above 15 mg/kg are considered high for agricultural purposes. Thus, the sampled Molopo soils are far below this value, possibly explaining the high acidity of the soil.

Cation leaching leading to acidity is common in sandy soils. Na uptake by plants helps to build pressure within cells (to sustain turgor). Similarities between K and Na ions can cause plants' ion transport pathways to struggle with differentiating between the two cations [94,95]. This can lead to excess Na cations being absorbed, which can become toxic to the plant and lead to K deficiencies [94]. Moreover, high Na concentrations can lead to reduced availability of other nutrients, such as Ca, as observed in this study (Table 2) [95].

## 5. Conclusions

Bush encroachment remains a serious problem in the semi-arid Molopo region of the North West and Northern Cape Provinces of South Africa. This study therefore aimed to disentangle the effects bush encroachment and the control thereof has on the functioning of landscapes in semi-arid rangelands such as these. The main conclusions of this study include:

1.  Bush-encroached sites can be considered as fully functional landscapes in drought years that will persist in their current state for many years if no control technologies were to be applied.

2.  An increase in shrub cover can be considered as a means to compensate for the loss of function that would, initially, have been supplied by grassy swards since shrubs have different modes of using soil moisture and cycling mineral nutrients and also have the ability to create more robust, long-lasting litter. In this way, despite the loss of herbaceous vegetation, the functioning of bush-encroached landscapes is largely retained but by different biophysical processes.

3.  It is also possible that shrubs may protect landscapes against much more serious degradation under drought conditions. The functionality of bush-encroached sites is further reinforced by the similarity between the scores derived for bush-controlled sites in this study. This, however, should be viewed cautiously, since sampling was conducted during an extreme drought event. Nutrient cycling, stability and infiltration—all indicators of the functionality of the landscape—did not differ between bush-controlled and bush-encroached sites. Long-term investigation is therefore proposed.

4.  Hand-controlled (HC) and hand-controlled with re-application (2HC) sites had the highest functionality scores for both survey years. These high functionality scores can be ascribed to the high amounts of litter and cryptogams found throughout the three different treatments. The shade created by larger grass litter and shrub litter patches found at these sites provided suitable sub-habitats for the cryptogams. The deep, loose, sandy soils of the Molopo region also played a role in the low stability and above-average infiltration scores recorded for all sites as well as the increase in deep-soil mineral nutrient cycling. Large inter-patches found between most of the nutrient accumulating patches played a vital role in similar functionality scores being recorded for bush-controlled and bush-encroached sites over both survey years. High grazing intensities and the lack of forage, as a result of the drought, could be the two main contributing factors that caused the increase in inter-patch size, as well as the decrease in the size and frequency of the nutrient accumulating patches.

Due to a lack of forage, land managers are forced to graze their livestock in bush-encroached and degraded areas during drought-stricken years. It is suggested that land managers take measures to promote soil nutrients in these large inter-patches to enhance their functioning. For example, higher nutrient content was observed when dead tree material and litter were left intact after control measures have been applied. Land managers should consider implementing similar strategies to enhance the overall functionality of their rangelands, even on the inter-patches irrespective of whether sites are controlled for bush encroachment or not, especially during higher rainfall periods. Furthermore, if land managers contemplate cutting shrubs manually, they should consider arranging the debris

into packs along the contour to act as resource transport regulators and to, perhaps, form seedling protection systems.

Given the similarity in nutrient levels recorded for bush-encroached and -controlled sites, our study suggests that controlling bush encroachment in the semi-arid rangelands of the Molopo does not affect soil nutrients. It is, however, important to investigate these patterns in terms of the different patches found in the landscape, since shrub litter patches of the bush-thickened sites and grass litter patches of the aeroplane-controlled sites had higher Ca, C and K concentrations compared to the patches of the other sites (i.e., all other bush-controlled sites: stem burning, sustainably managed, hand controlled and hand controlled with re-application). These high nutrient levels could be attributed to the nutrient deposition by the roots of the vegetation that occurred close to the soil surface and the increase in the quantity of litter, as well as the high densities of cryptogams found surrounding shrub litter patches and grass litter patches.

Even though bush encroachment is considered a problem in the semi-arid rangelands of the Molopo, our results suggest that this phenomenon does not have undue adverse effects on the functioning of ecosystems during periods of drought. However, when exploring the functioning of these rangelands, it is important to keep the distinct attributes of the study area in mind. For example, stability proved to be the most important factor with respect to the overall functioning of the system, which was to be expected since the Molopo is associated with dystrophic (i.e., nutrient-poor) sandy soils with high infiltration rates. Additionally, future research should focus on the long-term monitoring of ecosystem functionality within semi-arid rangelands with regards to bush control and the impacts thereof on the soil profile in a year with average or above-average rainfall.

**Author Contributions:** K.K. was responsible for the design and application of the project. K.K. also helped with the field work. J.F. analysed the data and results were used in the MSc thesis. D.T. helped with the description and application of the LFA methodology. R.B. assisted with interpretation and discussion of the soil analysis. K.K., J.F., D.T. and R.B. also helped with the editing of the manuscript. H.v.C. and N.v.S. assisted in the writing and editing of the manuscript and completed some of the statistical analyses. All authors have read and agreed to the published version of the manuscript.

**Funding:** This research was funded by the Unit for Environmental Sciences and Management (UESM) and Department of Forestry, Fisheries and the Environment (DFFE)—Bush Expert (Project financial number—3X00172).

**Institutional Review Board Statement:** Not applicable.

**Informed Consent Statement:** Not applicable.

**Data Availability Statement:** Data will be made available to all stakeholders.

**Acknowledgments:** This study formed part of a management and restoration sub-project (B2) resorting under a project titled "Integrative Decision-support System for Sustainable Rangeland Management in Southern African Savannas (IDESSA)" currently undertaken collaboratively by the NWU and the Universities of Goettingen, Marburg and KwaZulu-Natal. The project is funded by Germany's Federal Ministry of Education and Research (*Bundesministerium für Bildung und Forschung*). The research team would like to acknowledge Eco-Analytica (the laboratory housed at the North-West University's Potchefstroom campus in South Africa) that analysed the soil samples and the South African Weather Services (SAWS) (in particular the Severn and Bray weather stations) that provided the rainfall data. We would also like to thank Bianca Greyvenstein for her assistance and creation of the map in GIS.

**Conflicts of Interest:** The authors declare no conflict of interest. The funders had no role in the design of the study; in the collection, analyses, or interpretation of data; in the writing of the manuscript, or in the decision to publish the results.

# Appendix A

**Table A1.** Detailed description of selected study sites.

| Study Area Location | Farm 1: Bray (Figure 1)<br>Lat. (South): 25.38378<br>Long. (East): 23.38014 | Farm 2: Bray (Figure 1)<br>Lat. (South): 25.36234<br>Long. (East): 23.37437 | Farm 3: Vorstershoop (Figure 1)<br>Lat. (South): 25.81379<br>Long. (East): 22.8565 | Farm 4: Severn (Figure 1)<br>Lat. (South): 26.54924<br>Long. (East): 22.56828 |
|---|---|---|---|---|
| Total number of sampling sites | AC: 4 sites<br>BT: 4 sites | AC: 4 sites<br>BT: 4 sites<br>HC: 4 sites | BT: 3 sites<br>SB: 3 sites<br>SM: 3 sites | BT: 4 sites<br>HC: 4 sites<br>HC2: 4 sites |
| Land tenure type | Commercial game farming | Commercial cattle farming | Commercial cattle farming | Commercial cattle farming |
| Stock composition | Game:<br>● Giraffe (browser)<br>● Oryx (grazer)<br>● Impala (grazer / browser)<br>● Kudu (browser)<br>● Gnoe (grazer)<br>● Eland (grazer)<br>● Blesbok (grazer)<br>● Springbok (grazer)<br>● Buffalo (grazer)<br>● Sable antelope (grazer)<br>● Zebra (grazer) | Cattle and horses | Cattle | Cattle |
| Current stocking rate | 10 large stock units (LSU) per ha | 10 LSU/ha | 12 LSU/ha | 10 LSU/ha |
| Grazing system | Open system—no camps (paddocks) | Four-camp rotational grazing system rotating every two weeks | Eight-camp rotational grazing for nine days; resting period of 63 days per camp | Six-camp rotational grazing with two camps resting for an entire growing season |
| Chemical treatment with arboricide | Non-selective Tebuthiuron granules were applied in a grid formation by aeroplane (AC) at a dosage of 2.5–3 kg/ha to camps measuring 1000 ha. **The camps were treated in 2008/2009. Chemicals are reapplied every 10 years**. Reapplication falls outside this project's timeframe.<br>**Selected species:** *Senegalia mellifera, Vachellia luederitzii* | Non-selective (AC) and selective (HC). Tebuthiuron granules were applied by AC and by hand (HC) at a dosage of 3 kg/ha to camps measuring 245 ha. **The camps were treated in 2008/2009. Selected species:** *Dichrostachys cinerea, S. mellifera, Terminalia sericea V. luederitzii* | **Stem burning (SB) was first conducted in 1982. The second SB treatment was conducted in 1999/2000. Selected species:** *S. mellifera, V. luederitzii* | **The double hand-control sites (HC2) were treated in 2000 and followed up in 2012 at the start of the growing season (Oct./Sep.). Single hand-control sites (HC) were treated in 2006 (selective).** Tebuthiuron granules were used. The treated HC2 camps measured 240 ha and 260 ha, and the HC camps measured 119 ha and 115 ha. **Species selected:** *Rhigozum trichotomum, S. mellifera, V. luederitzii* |

## Appendix B. Description of Patches

*Appendix B.1. Inter-Patches (Ips)*

An inter-patch is classified as any patch under the line of the transect with no rooted perennial plant material present that might obstruct nutrients or water (resources) from flowing out of the system [96]. Inter-patches can further be described as zones where water, soil particles and litter are free to move either in the direction the wind is blowing or downslope if the velocity of rainwater is high enough to transport or move litter [96]. Inter-patches were the most dominant patch type across bush-controlled treatments in this study.

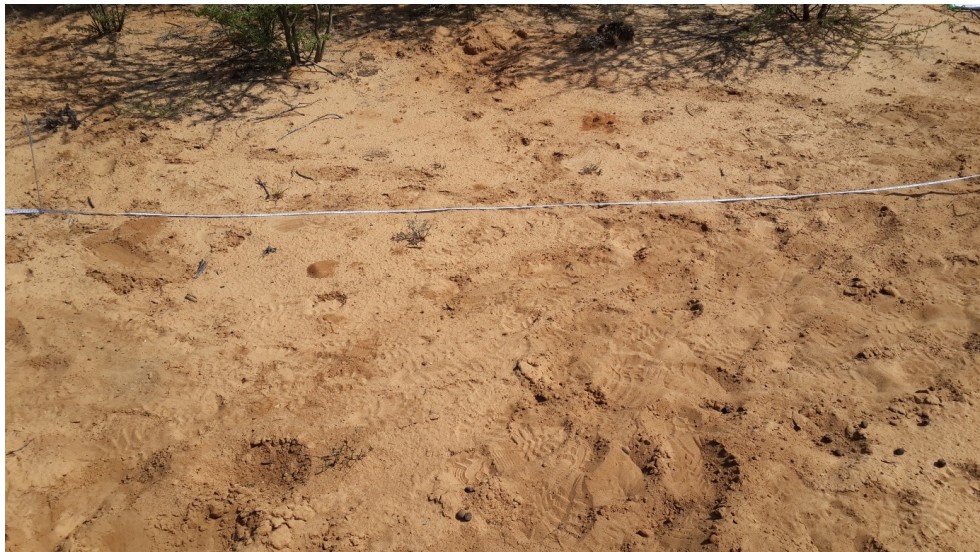

**Figure A1.** A typical inter-patch displaying exposed bare soil.

*Appendix B.2. Grass Patches (GPs)*

These patches contain rooted perennial grass species. Annual grass tufts are not included in GPs since they do not make a significant contribution to landscape stability, infiltration and nutrient cycling [96]. A GP can either be a single tuft of grass or a combination of perennial grasses dominant in the landscape forming a unit where resources are trapped. Due to the drought, GPs identified during this study tended to be either heavily grazed in the encroached site (if the grass species was palatable) or extremely dry (if unpalatable). Bush-controlled sites were associated with larger GPs.

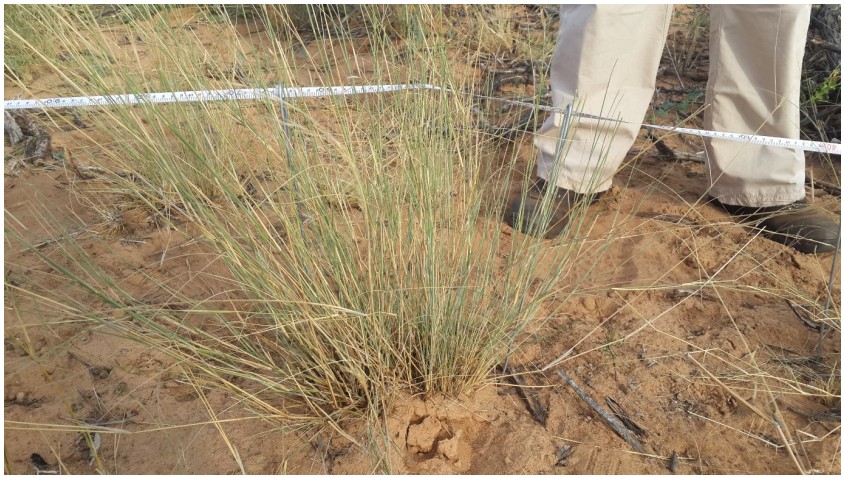

**Figure A2.** Example of a grass patch prevalent in controlled areas.

*Appendix B.3. Grass Litter Patches (GLPs)*

A grass litter patch (GLP) is a combination of plant debris composed of material such as dead branches and dead grass patches (litter) and any perennial grass tufts (actively growing). Both grass tufts and tree branches act as resource-retaining zones. The resources trapped by these patches serve as a means to enrich the soil surrounding the patch [96].

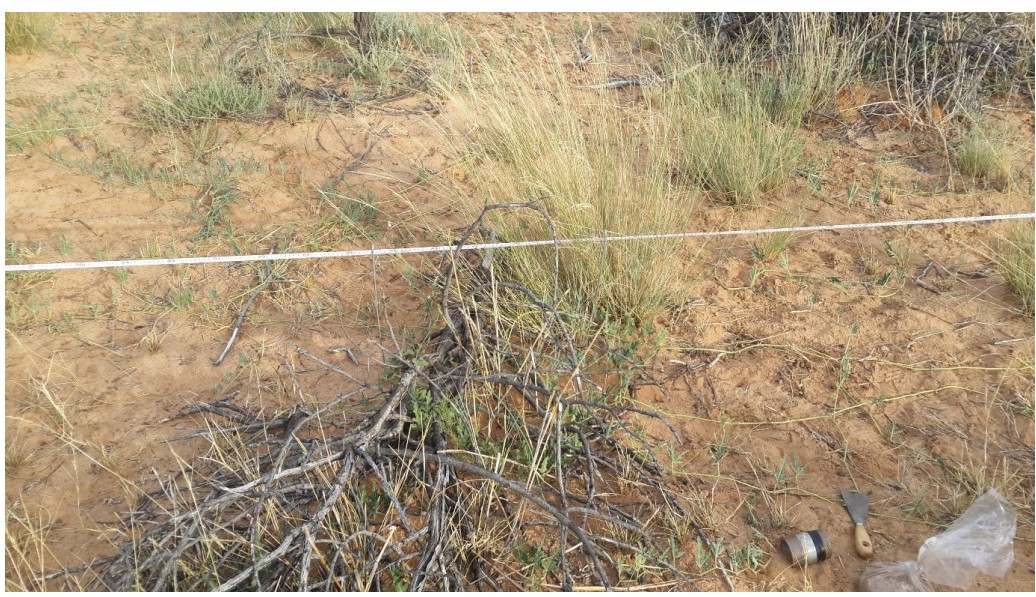

**Figure A3.** Typical grass litter patch.

*Appendix B.4. Litter Patches (LPs)*

LPs act as fertile islands that aid the establishment of new tree seedlings or grass tufts by trapping resources, especially seed, water and nutrients [96]. Mostly composed of woody biomass and/or litter stemming from bush-clearing exercises, this study showed a lot of variation in the composition of these patches (density and arrangement of litter) but found these patches to be much larger at controlled sites compared to those found at encroached sites.

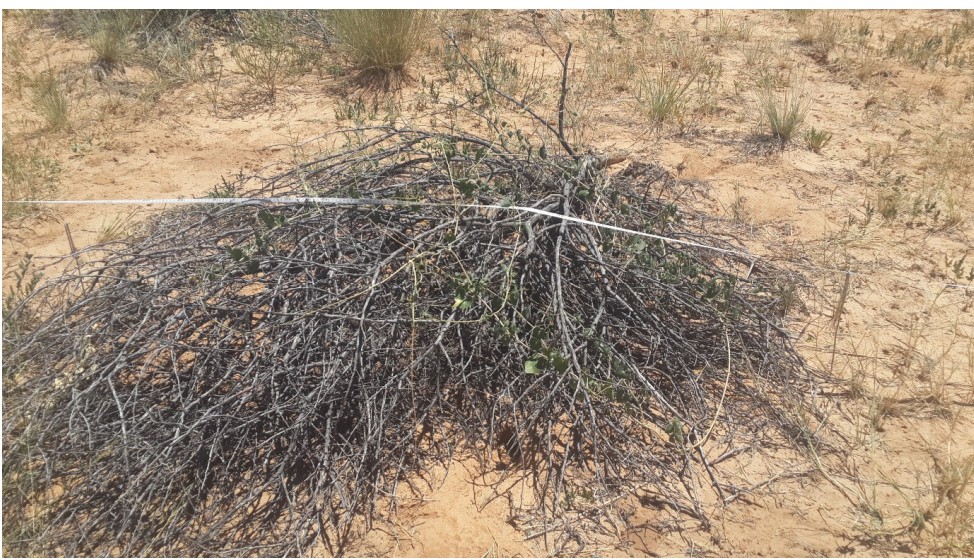

**Figure A4.** An example of a large litter patch in a controlled site.

*Appendix B.5. Shrub Patches (SPs)*

Shrub patches (SPs) are composed of small shrubs with a height of 30 cm and above to taller trees of 2 m and above where both the shrubs and the trees have one or more stem, irrespective of their height class. In this study, shrub patches (SPs) were commonly found in encroached sites and were characterised by woody trees and/or shrubs that contribute to bush encroachment, such as *Dichrostachys cinerea*, *Senegalia mellifera* and *Vachellia luederitzii*. SPs were, however, also identified in controlled and sustainably managed sites but differed in species composition and were mostly composed of *Boscia albitrunca*, *Vachellia erioloba* and *Grewia flava*.

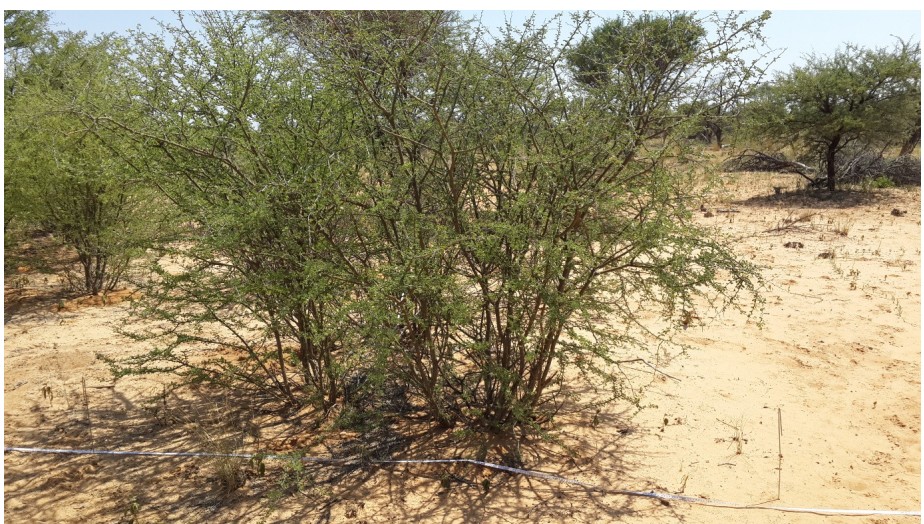

**Figure A5.** A typical shrub patch (SP) characterised by living trees/shrub.

*Appendix B.6. Shrub Litter Patches (SLPs)*

Whenever litter is detected beneath an SP patch, that patch is classified as a shrub litter patch (SLP). In this study, litter beneath SPs varied considerably, ranging from dead grass to small branches. Patches such as these were common to encroached sites and were also found within controlled and sustainably managed sites.

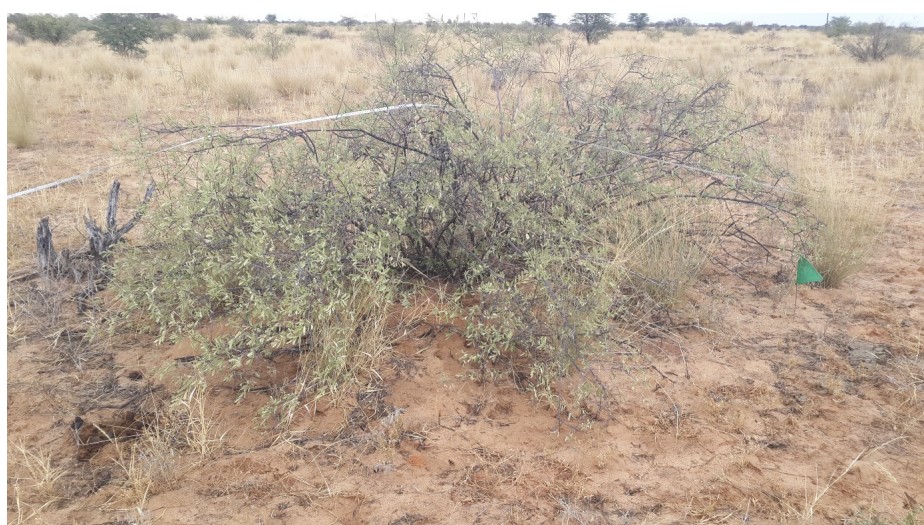

**Figure A6.** A typical shrub litter patch, displaying both living woody shrub and litter.

*Appendix B.7. Grass Shrub Patches (GSPs)*

Grass shrub patches (GSPs) are characterised by woody shrubs that, amongst others, act as a defence mechanism to protect the grass growing underneath against herbivory and high temperatures. In this study, shrubs providing this type of protection mostly belonged to the *Grewia* spp., with *Grewia flava* being the dominant shrub in sustainably managed and bush-controlled sites.

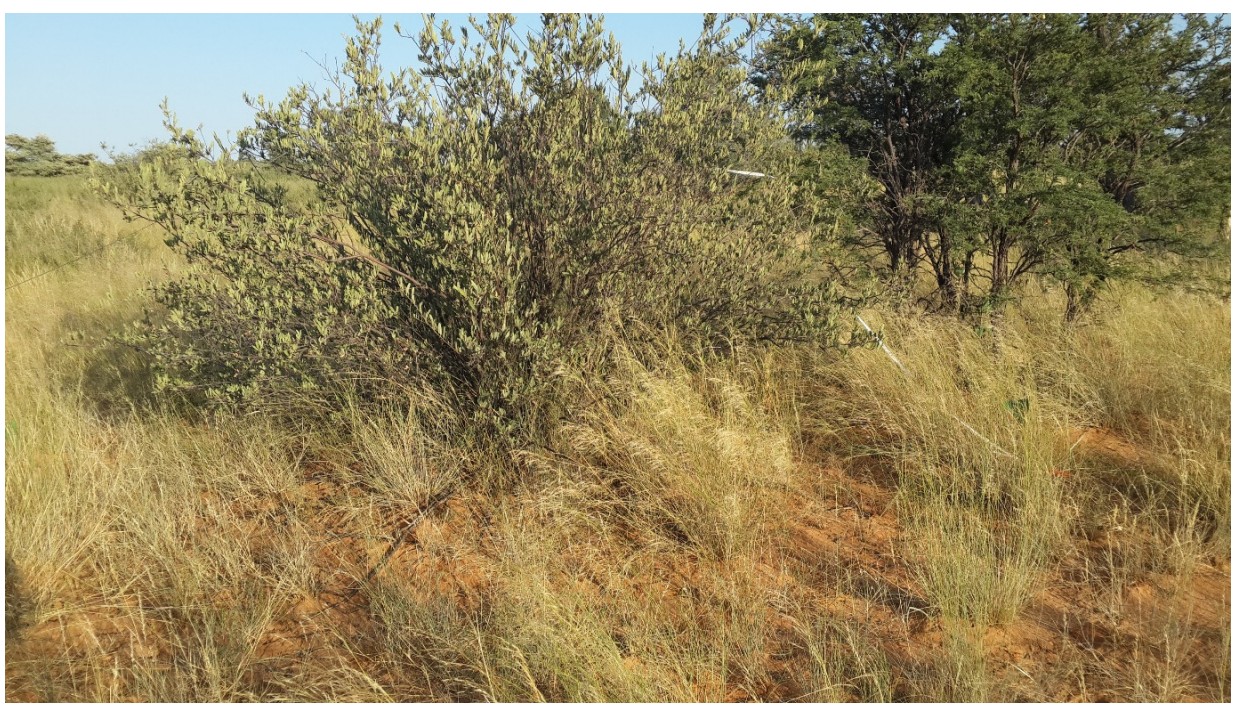

**Figure A7.** A typical grass shrub patch. All photographs courtesy of JH Fouché.

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
