# Peer review of "Landscape Function Analysis: Responses to Bush Encroachment in a Semi-Arid Savanna in the Molopo Region, South Africa"

_sustainability, doi:10.3390/su14148616_

Round 1

Reviewer 1 Report

I think that the paper was written in accordance with the instructions for writing papers in the journal Sustainabilitj. The paper deals with specific issues that are not very common, especially if we keep in mind the object of research.

The paper is based on good statistical processing methods and the results are presented correctly.

Author Response

Please see new, corrected manuscript and rebuttal in attached file.

Reviewer 2 Report

Various factors lead to increased shrub density, biomass and cover (so-called ‘bush encroachment’) that influence ecosystem functioning and services in semi-arid rangelands. Ultimately, bush encroachment has adverse effects on human livelihoods. An increased understanding of ecosystem functioning in bush-encroached rangelands could contribute to improved management, conservation and restoration. This study, therefore, aimed to determine landscape functioning of bush-encroached and controlled savanna rangelands in the Molopo region, South Africa, by using the landscape function analysis (LFA) monitoring procedure. Basically, the manuscript is well organized and well written. However, I have some editorial and technical comments, as listed below, to the authors for further improving the quality of this paper.

(1)Some more relative references should be added and cited in the section of Introduction.

The Changes of Spatiotemporal Pattern of Rocky Desertification and Its Dominant Driving Factors in Typical Karst Mountainous Areas under the Background of Global Change(https://doi.org/10.3390/rs14102351)

The dominant influencing factors of desertification changes in the source region of Yellow River Climate change or human activity?(http://dx.doi.org/10.1016/j.scitotenv.2021.152512)

(2)Please improve the quality of Figure 4.

(3)  Please rearranged the section of results and discussions, separately.

(4)The section of conclusion was not concise and sub-section of conclusion should be re-arranged as (1)(2)…(5).

Author Response

(The authors gave the same response as above.)

Reviewer 3 Report

The research provides an initial research on the impacts of shifts in vegetation composition and encroachment that can affect the eco-system in savanna landscapes. Comments to consider:

·      - Would be highly beneficial the provide photos of savanna’s vegetation communities that are discussed in the paper (to get sense of the landscape and the issue).

·       -  LFA to be clearly explained. There’re many functions to be studied but what is the focus of this method to explained as a separate heading.

·       -  What is the rate and size of bush encroachment?

·      -   What are the benefits of bush encroachment for increasing climate change resilience?

Author Response

(The authors gave the same response as above.)
